# Multimodal Fusion of RGB and Complementary Modalities for Semantic Segmentation

## Abstract

Multi-modal semantic segmentation augments RGB imagery with an auxiliary sensing stream X (RGB+X)—such as thermal, LiDAR, event, polarization, or light field—to enhance robustness under adverse illumination and motion blur. However, sensor heterogeneity often leads to misaligned features and unstable fusion. To alleviate these issues, we propose a bidirectional polarity-aware cross-modality fusion (BPCF) module that effectively captures complementary cues while enhancing feature alignment. We evaluate the framework on five modality pairings—RGB+Thermal, RGB+LiDAR, RGB+Event, RGB+Polarization, and RGB+Light Field—and achieve state-of-the-art results on eight public datasets. Notably, our method delivers a **17%** mIoU absolute improvement over the second-best approach on the MFNet dataset.

## 1 Introduction

In autonomous driving and robotic perception, RGB-only semantic segmentation often suffers from performance degradation under low illumination, fast motion, and adverse weather conditions Bijelic et al. (2020). Non-RGB modalities provide complementary cues that can mitigate these limitations. For instance, incorporating thermal, LiDAR, event, light field, and polarization data can enhance robustness against low-light conditions, motion blur, and weather-induced artifacts Chen et al. (2020a); Bijelic et al. (2020). Nevertheless, effectively fusing an increasing number of modalities in a reliable manner—while fully leveraging the strengths of each sensor—remains an open challenge.

This problem can be considered from two perspectives. The first is the efficient utilization of complementary information. In multi-modal scenarios such as RGB–Thermal or RGB–Event, the same object may exhibit reversed attention responses across modalities.

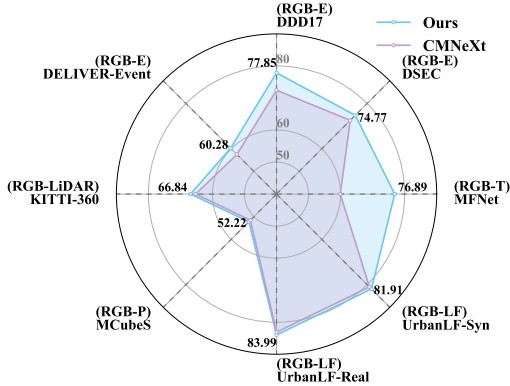

Figure 1: Performance comparison between our proposed method (Ours) and CMNeXt Zhang et al. (2023) across different modality combinations, including RGB–Event, RGB–Thermal, RGB–Light Field, RGB–LiDAR, and RGB–Polarization.

Recognizing that such negative correlations can also provide valuable cues, we propose a bidirectional polarity-aware linear cross-attention (BPLCA) mechanism. BPLCA decomposes features into positive and negative components and performs cross-modal attention across both branches. This design enables the model to capture not only shared but also complementary signals across modalities, leading to more complete and balanced feature integration.

The second perspective is cross-modal feature alignment. Heterogeneous data sources (e.g., RGB, thermal, LiDAR, events, and polarization) exhibit distinct statistics and styles, where naive fusion often leads to inconsistent features or an over-reliance on a single modality Liu et al. (2022); Zhang et al. (2023); Cao et al. (2024). To alleviate this issue, we introduce the dual feature consistency con-

straint (DFCC), which aligns cross-modal features and promotes coherent semantic representations across modalities. By integrating the above designs, our framework achieves superior performance compared to recent state-of-the-art (SOTA) methods across eight multi-modal datasets, as illustrated in Fig. 1. The main contributions of this work are as follows:

- We propose a multi-modal semantic segmentation framework that supports flexible modality combinations. It enables effective representation learning across diverse RGB+X configurations, including RGB+Thermal, RGB+LiDAR, RGB+Event, RGB+Polarization, and RGB+Light Field.

- We develop a bidirectional polarity-aware cross-modality fusion (BPCF) module, which integrates bidirectional polarity-aware linear cross-attention (BPLCA) with a dual feature consistency constraint (DFCC) to effectively fuse heterogeneous modalities.

- We introduce a stage-wise constraint loss that progressively enforces consistent cross-modal feature alignment.

- We conduct extensive experiments on eight multi-modal datasets covering five auxiliary modalities. Our method consistently outperforms SOTA approaches across all datasets.

## 2 RELATED WORK

Recognizing the auxiliary benefits of non-RGB modalities, prior work has explored a spectrum of fusion strategies, ranging from pairwise integration of complementary modalities to modality-agnostic frameworks capable of unifying many sensor types. Dual-branch backbones integrate events over time and use event-count or activity-rate cues, together with bidirectional cross-attention, to synchronize features across modalities and scales Sun et al. (2021); Zhou et al. (2023); Xie et al. (2024); Zheng et al. (2024a); Li et al. (2025a). Beyond RGB-event fusion, other auxiliary modalities have also been explored Prakash et al. (2021); Joze et al. (2020); Hazirbas et al. (2016). Moreover, KTBNet Cai et al. (2025) proposes a parameter-efficient symmetric framework that balances the contributions between RGB and an additional modality such as thermal, thereby preventing the dominance of a single branch. Polarization offers informative cues, being particularly sensitive to specular reflections and material or surface boundaries. However, prior work on RGB–P segmentation has shown only limited performance Liang et al. (2022); Zhang et al. (2023). In parallel, RoadFormer Huang et al. (2024) integrates RGB and polarization via a dual-branch fusion block and achieves commendable performance. Nevertheless, the above designs lack modality-agnostic generalization, as they are tied to fixed modality combinations. CMX Liu et al. (2022) employs cross-modal feature rectification and fusion modules for long-range context exchange across modalities. Its successor CMNeXt Zhang et al. (2023) extends scalability through a self-query hub and a parallel pooling mixer, maintaining compactness while adapting to diverse sensor types. Similarly, Zheng et al. Zheng et al. (2024b) propose a modality-agnostic pipeline that dynamically selects the most informative modalities, while their follow-up Zheng et al. (2024a) introduces a modality-agnostic Feature Fusion (MFF) module that synthesizes heterogeneous sensor streams into a unified representation. GeminiFusion Jia et al. (2024) further refines cross-modal interactions using pixel-wise intra- and inter-modal attention, and OmniVec2 Srivastava & Sharma (2024) explores a large-scale shared representation space for multi-modal and multi-task learning. Although these works have advanced the field, challenges still remain in fully exploiting complementary information across modalities and in achieving consistent feature alignment. In this work, we propose a bidirectional polarity-aware cross-modality fusion (BPCF) module to alleviate these issues.

## 3 METHOD

As shown in Fig. 2, our model adopts an encoder–decoder architecture with stage-wise fusion modules. Each modality is first encoded by a dedicated Transformer backbone Xie et al. (2021). Following prior work Liu et al. (2022); Zhang et al. (2023), we incorporate ScoreNet to flexibly integrate RGB with one or more auxiliary modalities. For multiple auxiliary inputs, ScoreNet selects the most informative feature at each spatial location. The stage-wise BPCF fusion modules then combine the hierarchical feature representations from the two modality branches. Finally, the fused multi-scale features are aggregated by an MLP-based segmentation head to produce dense predictions.

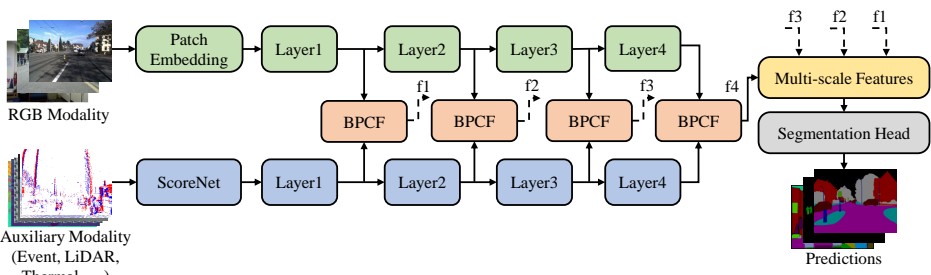

Figure 2: Overview of the proposed multi-modal semantic segmentation framework. Each modality is independently encoded using a dedicated Transformer backbone. RGB and auxiliary modality features are then progressively fused through the proposed fusion module in a stage-by-stage manner. Finally, the fused multi-scale features are passed to the segmentation head to generate predictions.

## 3.1 FEATURE EXTRACTION

**ScoreNet.** To extract the most informative modality features, we employ ScoreNet with a dynamic selection mechanism. Specifically, ScoreNet processes each patch in each auxiliary modality as follows:

$$S_{i,j} = \text{Softmax}(\text{Linear}(\text{GELU}(\text{Linear}(\text{Norm}(X_{i,j}))))),\ i \in [1, M],\ j \in [1, N]. \tag{1}$$

where $S_{i,j}$ denotes the informative score for the $i$-th input modality on the $j$-th patch token, $M$ refers to the number of modalities and $N = H \times W$ refers to the number of patches. For each patch token, the embedding from the modality with the highest score is dynamically selected and fused into a single auxiliary input to complement the primary RGB data:

$$x_{\text{a}} = \{X_{m,j} | j \in H \times W, m = \underset{i}{\text{argmax}}(S_{i,j})\}. \tag{2}$$

**Backbone.** Subsequently, the primary RGB data and the dynamically selected auxiliary modality are encoded by parallel Transformer backbones (MiT) Xie et al. (2021). The extracted multi-level features are then passed to the proposed stage-wise fusion moldues. The process can be formulated as follows:

$$x_{\text{r}} = \text{MiT}(x_{\text{r}}),\ x_{\text{a}} = \text{MiT}(x_{\text{a}}). \tag{3}$$

## 3.2 BIDIRECTIONAL POLARITY-AWARE CROSS-MODALITY FUSION MODULE

To effectively integrate heterogeneous cues from RGB and auxiliary modalities, we introduce a bidirectional polarity-aware cross-modality fusion (BPCF) module, as illustrated in Fig. 3. The BPCF is composed of two key parts: bidirectional polarity-aware linear cross-attention (BPLCA) and dual feature consistency constraint (DFCC). The BPLCA facilitates comprehensive cross-modal interaction by employing symmetric cross-gating and polarity-aware cross-linear attention. Meanwhile, the DFCC enhances feature alignment by exploiting statistical correlations and applying the consistency constraint loss $\mathcal{L}_{\text{stage}}$, complemented by a refinement block to generate the final fused output.

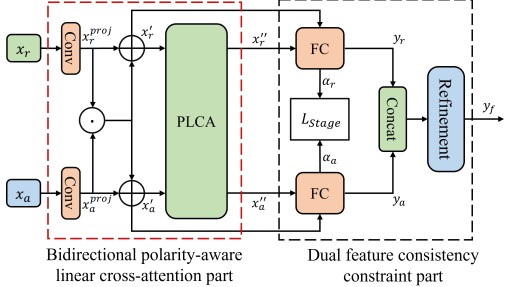

Figure 3: The overall structure of the BPCF module. It consists of a bidirectional polarity-aware linear cross-attention (BPLCA) part and a dual feature consistency constraint (DFCC) part.

**Bidirectional polarity-aware linear cross-attention (BPLCA).** To comprehensively fuse complementary cues from the RGB stream and the auxiliary modality stream, our BPLCA is designed with

dual cross-modal fusion branches. First, an efficient cross-gating mechanism is applied to activate relevant features. Second, a symmetric polarity-aware linear cross-attention (PLCA) is introduced (see Fig. 4), which preserves informative negative evidence while maintaining the linear-time and linear-memory efficiency of kernelized attention. Formally, let $x_r, x_a \in \mathbb{R}^{N \times C}$ represent the extracted features from the RGB and auxiliary branches, reshaped into sequences of $N = HW$ tokens with $C$ channels. Two $1 \times 1$ convolutions are then used to perform per-location linear projections:

$$x_r^{\text{proj}} = \text{Conv}_{1 \times 1}(x_r), x_a^{\text{proj}} = \text{Conv}_{1 \times 1}(x_a). \tag{4}$$

We then realize bidirectional interaction via symmetric cross-gating using the element-wise (Hadamard) product $\circ$:

$$x_r' = x_r^{\text{proj}} + (x_r^{\text{proj}} \circ x_a^{\text{proj}}), x_a' = x_a^{\text{proj}} + (x_r^{\text{proj}} \circ x_a^{\text{proj}}). \tag{5}$$

To further enhance cross-modal interaction, the fused features are passed into the PLCA, as illustrated in Fig. 4. Within PLCA, the input features are projected into four distinct matrices: query $(Q)$, key $(K)$, value $(V)$, and gating $(G)$. Among these, the gating matrix $G$ plays a pivotal role in realizing polarity-aware attention Meng et al. (2025), whose mechanism will be detailed in the following section.

$$Q_r = x_r' q_r, \ K_r = x_r' k_r, \ V_r = x_r' v_r, \ G_r = x_r' g_r;$$
$$Q_a = x_a' q_a, \ K_a = x_a' k_a, \ V_a = x_a' v_a, \ G_a = x_a' g_a. \tag{6}$$

where $q_r, k_r, v_r, g_r, q_a, k_a, v_a, g_a$ denote the learnable linear projections for the RGB and auxiliary branches, respectively.

Following the kernelization strategy in Katharopoulos et al. (2020), vanilla attention can be reformulated using a similarity function defined by a feature map $\phi(\cdot)$. The $i$-th row of the attention result is formulated as:

$$\text{Attention}(Q, K, V)_i = \frac{\sum_j \text{sim}(Q_i, K_j) V_j}{\sum_j \text{sim}(Q_i, K_j)}, \tag{7}$$

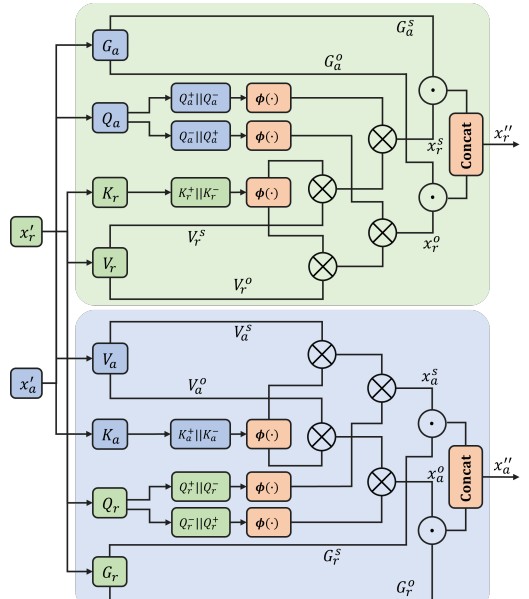

Figure 4: Illustration of PLCA. Features from RGB and auxiliary modality are deeply fused through the PLCA component.

$$\text{sim}(x, y) = \kappa(x, y) = \phi(x)^T \phi(y). \tag{8}$$

This kernelization renders the similarity function decomposable. By exploiting distributive and associative properties, the final attention computation becomes:

$$\text{Attention}(Q, K, V) = \frac{\phi(Q)^T \left( \sum_j \phi(K_j) V_j \right)}{\phi(Q)^T \left( \sum_j \phi(K_j) \right)}. \tag{9}$$

Notably, Eq. 9 avoids the $\mathcal{O}(N^2 D)$ pairwise computations $Q_i^\top K_j$ by precomputing the terms $\sum_j \phi(K_j) V_j$ and $\sum_j \phi(K_j)$, thereby reducing the overall time complexity to $\mathcal{O}(N D^2)$. Standard non-negative feature maps like ReLU discard negative information. This information is retained by decomposing queries and keys into positive $(Q^+)$ and negative $(Q^-)$ parts via $Q = Q^+ - Q^-$ where $Q^+ = \text{ReLU}(Q)$ and $Q^- = \text{ReLU}(-Q)$. The same decomposition is applied to the key matrix $K$. Then the inner product can be decomposed into:

$$\langle q_i, k_j \rangle = \langle q_i^+, k_j^+ \rangle + \langle q_i^-, k_j^- \rangle - \langle q_i^-, k_j^+ \rangle - \langle q_i^+, k_j^- \rangle. \tag{10}$$

Then the linear-decomposable similarity function in Eq. 8 can be derived as

$$
\begin{aligned}
\text{sim}(x, y) &= \phi(q_i)^T \phi(k_j) \\
&= ((\phi(q_i)^+)^T \phi(k_j)^+ + (\phi(q_i)^-)^T \phi(k_j)^-) - ((\phi(q_i)^+)^T \phi(k_j)^- + (\phi(q_i)^-)^T \phi(k_j)^+).
\end{aligned}
\tag{11}
$$

Thus, we can accept a non-positive feature map $\phi$ in the kernel. To process these non-negative components, we employ a channel-wise learnable feature map, $\phi(x)$, defined as:

$$
\phi(x) = x^P, \text{where } P = 1 + \alpha \cdot \text{sigmoid}(W)
\tag{12}
$$

where $P$ is a dynamically computed exponent, determined by a hyperparameter $\alpha$ and a $d$-dimensional learnable parameter vector $W$, which learns to weigh the relative importance of different channels. To explicitly model the interactions between polarities, we first concatenate the positive and negative components along the channel dimension (denoted by $\|$), then capture the interactions between $Q$ and $K$ for *same-signed* $(++, --)$ and *opposite-signed* $(+-, -+)$ pairs separately while ensuring all kernelized terms remain non-negative. The value $(V)$ is partitioned accordingly with $V_r = V_r^s \| V_r^o$ and $V_a = V_a^s \| V_a^o$, while both split tensors have half of the channels from the original one. Based on these decompositions, we then compute the intermediate fusion results for the RGB and auxiliary modalities, distinguishing between *same-signed* and *opposite-signed* correlations, substituting Eq. 11 into Eq. 9:

$$
x_r^s = \frac{\phi(Q_a^+ \| Q_a^-)^T \sum_j \phi(K_r^+ \| K_r^-)_j (V_r^s)_j}{\phi(Q_a^+ \| Q_a^-)^T \sum_j \phi(K_r^+ \| K_r^-)_j}, x_r^o = \frac{\phi(Q_a^- \| Q_a^+)^T \sum_j \phi(K_r^+ \| K_r^-)_j (V_r^o)_j}{\phi(Q_a^- \| Q_a^+)^T \sum_j \phi(K_r^+ \| K_r^-)_j};
$$

$$
x_a^s = \frac{\phi(Q_r^+ \| Q_r^-)^T \sum_j \phi(K_a^+ \| K_a^-)_j (V_a^s)_j}{\phi(Q_r^+ \| Q_r^-)^T \sum_j \phi(K_a^+ \| K_a^-)_j}, x_a^o = \frac{\phi(Q_r^- \| Q_r^+)^T \sum_j \phi(K_a^+ \| K_a^-)_j (V_a^o)_j}{\phi(Q_r^- \| Q_r^+)^T \sum_j \phi(K_a^+ \| K_a^-)_j}.
\tag{13}
$$

To capture sophisticated relations between *same-signed* and *opposite-signed* parts, we employ a learnable, element-wise mixing that *weights* the *same-signed* and *opposite-signed* streams via gating tensors, $G = G^s \| G^o$. The final polarity-aware outputs are then produced through gated fusion:

$$
x_r'' = (x_r^s \circ G_a^s) \| (x_r^o \circ G_a^o), x_a'' = (x_a^s \circ G_r^s) \| (x_a^o \circ G_r^o)
\tag{14}
$$

Notably, the query matrix $Q$ and gating matrix $G$ are designed to interact with the other modality in the *cross-modality* design of the PLCA component. This mechanism preserves the integrity of the original content alignment by mitigating excessive influence from the complementary modality, yet still allows for the infusion of necessary cross-modal information.

**Dual feature consistency constraint (DFCC).** To strengthen cross-modal feature alignment while maintaining consistency, we introduce the DFCC. As shown in Fig. 5, DFCC integrates a feature consistency constraint with a refinement block in a unified design, jointly enabling semantic alignment and feature enhancement. Specifically, given enhanced RGB and auxiliary features $x_r'$, $x_a' \in \mathbb{R}^{B \times C \times H \times W}$ and fused features $x_r''$, $x_a'' \in \mathbb{R}^{B \times C \times H \times W}$, we first compute feature statistics to align the feature spaces across both streams:

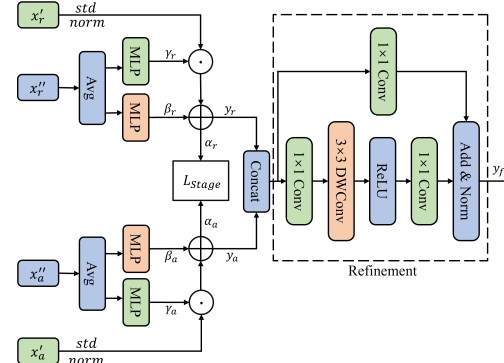

Figure 5: Illustration of DFCC.

$$
\hat{x}_r = \frac{x_r' - \mu(x_r')}{\sigma(x_r')}, \hat{x}_a = \frac{x_a' - \mu(x_a')}{\sigma(x_a')};
\tag{15}
$$

where,

$$
\mu(x) = \frac{1}{HW} \sum_{h=1}^{H} \sum_{w=1}^{W} x, \sigma(x) = \sqrt{\frac{1}{HW} \sum_{h=1}^{H} \sum_{w=1}^{W} (x - \mu(x))^2}.
\tag{16}
$$

where $\mu(x)$ and $\sigma(x)$ denote the channel-wise mean and standard deviation, respectively. DFCC then learns adaptive affine modulation parameters from the fused features $x''_r$ and $x''_a$. Specifically, the global average pooling operations are first applied:

$$z_r = \text{AvgPool}(x''_r), z_a = \text{AvgPool}(x''_a). \tag{17}$$

Subsequently, MLPs are employed to generate branch-specific modulation parameters:

$$\begin{aligned}
\gamma_r &= \text{MLP}(z_r), \beta_r = \text{MLP}(z_r), \\
\gamma_a &= \text{MLP}(z_a), \beta_a = \text{MLP}(z_a).
\end{aligned} \tag{18}$$

The normalized features are then modulated as:

$$y_r = \gamma_r \cdot \hat{x}_r + \beta_r, y_a = \gamma_a \cdot \hat{x}_a + \beta_a. \tag{19}$$

Finally, the two streams are concatenated and fed into the refinement block, which comprises two parallel paths: a $1 \times 1$ convolutional shortcut that preserves identity information, and a nonlinear transformation branch that enhances feature fusion. The nonlinear branch sequentially applies a $1 \times 1$ convolution for channel reduction, a $3 \times 3$ depthwise convolution for spatial interaction, a ReLU activation, and another $1 \times 1$ convolution to restore the channel dimension, followed by normalization. The final output is then obtained by summing the two paths:

$$y_f = \text{Refinement}(y_r \parallel y_a). \tag{20}$$

Moreover, to further enhance cross-modal consistency, we incorporate a stage-wise constraint loss within DFCC. At each fusion stage, the features $y_r$ and $y_a$ are computed by averaging over the channel dimension:

$$\begin{aligned}
\alpha_r &= \text{Norm}(\text{Mean}(y_r, \dim = 1)), \\
\alpha_a &= \text{Norm}(\text{Mean}(y_a, \dim = 1)).
\end{aligned} \tag{21}$$

where $\text{Norm}(a) = \frac{a - \min(a)}{\max(a) - \min(a) + \epsilon}$ is applied pixel-wise. The stage-wise constraint loss is then defined as the mean squared error (MSE) between the normalized features:

$$\mathcal{L}_{\text{stage}} = \|\alpha_r - \alpha_a\|_2^2. \tag{22}$$

This objective encourages both modalities to produce consistent responses, thereby improving feature alignment.

## 3.3 Loss Function

Our proposed model is trained using a combination of segmentation and alignment objectives. For the primary segmentation task, we employ the standard pixel-wise cross-entropy loss, commonly used in dense prediction tasks:

$$\mathcal{L}_{\text{CE}} = -\frac{1}{N} \sum_{i=1}^{N} \log \frac{\exp(p_{i,y_i})}{\sum_{j=1}^{C} \exp(p_{i,j})}. \tag{23}$$

where $N = H \times W$ denotes the number of valid pixels, $C$ is the number of classes, $p_{i,j}$ represents the logit for class $j$ at pixel $i$, and $y_i$ is the corresponding ground-truth label. The overall training objective combines the segmentation loss with the stage-wise constraint loss as follows:

$$\mathcal{L}_{\text{total}} = \lambda_1 \cdot \mathcal{L}_{\text{CE}} + \lambda_2 \cdot \sum_{i=1}^{4} \mathcal{L}_{\text{stage}}^{(i)}. \tag{24}$$

where $\lambda_1$ and $\lambda_2$ balance the segmentation and stage-level constraint losses. In our experiments, we set $\lambda_1 = 1$ and $\lambda_2 = 0.1$.

## 4 Comparison with State-of-the-arts

We evaluate our approach on eight widely used multi-modal datasets: MFNet Ha et al. (2017), KITTI-360 Liao et al. (2022), DDD17 Binas et al. (2017), DSEC Gehrig et al. (2021), DE-LIVER Zhang et al. (2023), UrbanLF Sheng et al. (2022), MCubeS Liang et al. (2022), and ZJU Xiang et al. (2021). Additional dataset details are provided in the **Appendix**.

Table 1: Semantic segmentation results on the MFNet RGB-Thermal dataset.

| Method | Venue | Backbone | mIoU (%) |
|---|---|---|---|
| PAP Zhang et al. (2019) | CVPR'19 | ResNet-18 | 50.5 |
| ABMDRNet Zhang et al. (2021) | CVPR'21 | ResNet-18 | 54.8 |
| GMNet Zhou et al. (2021b) | TIP'21 | ResNet-50 | 57.3 |
| EGFNet Zhou et al. (2022) | AAAI'22 | ResNet-152 | 54.8 |
| DooDLeNet Frigo et al. (2022) | CVPR'22 | ResNet-101 | 57.3 |
| CMX Liu et al. (2022) | TITS'23 | MiT-B4 | 59.7 |
| CMNeXt Zhang et al. (2023) | CVPR'23 | MiT-B4 | 59.9 |
| KTBNet Cai et al. (2025) | CVPR'25 | Swin-B | 59.9 |
| Ours | - | MiT-B4 | **76.9** |

Table 2: Semantic segmentation results on the KITTI-360 RGB-LiDAR dataset.

| Method | Venue | Backbone | mIoU (%) |
|---|---|---|---|
| PMF Zhuang et al. (2021) | ICCV21 | SalsaNext | 54.5 |
| TransFuser Prakash et al. (2021) | CVPR21 | RegNetY | 56.6 |
| TokenFusion Wang et al. (2022) | CVPR22 | MiT-B2 | 54.6 |
| HRFuser Broedermann et al. (2023) | ITSC23 | HRFormer-T | 48.7 |
| CMX Liu et al. (2022) | TITS23 | MiT-B2 | 64.3 |
| CMNeXt Zhang et al. (2023) | CVPR23 | MiT-B2 | 65.3 |
| Ours | - | MiT-B2 | **66.8** |

Table 3: Semantic segmentation results on the DDD17 and DSEC RGB-Event datasets.

| Method | Venue | Modal | DDD17 | | DSEC | |
|---|---|---|---|---|---|---|
| | | | mIoU (%) | Acc. (%) | mIoU (%) | Acc. (%) |
| E2VID Rebecq et al. (2019) | TRAMI'19 | Event | 48.47 | 85.84 | 44.08 | 80.06 |
| OpenESS Kong et al. (2024) | CVPR'24 | Event | 63.00 | 91.05 | 57.21 | 90.21 |
| KWYAF Li et al. (2025b) | AAAI'25 | Event | 57.69 | 90.04 | 57.75 | 90.87 |
| ESEG-L Zhao et al. (2025) | AAAI'25 | Event | 59.97 | 90.68 | 57.55 | 91.47 |
| CMX Liu et al. (2022) | TITS'23 | RGB+Event | 71.88 | 95.64 | 72.42 | 95.07 |
| CMNeXt Zhang et al. (2023) | CVPR'23 | RGB+Event | 72.67 | 95.74 | 72.54 | 95.10 |
| HMNet-L Hamaguchi et al. (2023) | CVPR'23 | RGB+Event | - | - | 55.00 | 89.80 |
| EISNet Xie et al. (2024) | TMM'24 | RGB+Event | 75.03 | 96.04 | 73.07 | 95.12 |
| Hybrid-Segmentation Li et al. (2025a) | AAAI'25 | RGB+Event | 67.31 | 95.07 | 66.57 | 94.27 |
| Ours | - | RGB+Event | **77.85** | **97.10** | **74.77** | **95.60** |

**Results on MFNet.** We evaluate our method on the MFNet dataset, a popular benchmark for RGB-thermal semantic segmentation. As shown in Tab. 1, our model achieves 76.9% mIoU, significantly outperforming prior methods such as KTBNet Cai et al. (2025) (59.9%) and CMNeXt Zhang et al. (2023) (59.9%). In particular, our approach yields a **17%** absolute improvement over KTBNet, highlighting its superior capability for RGB-thermal segmentation.The proposed method establishes a new SOTA result on the MFNet dataset.

**Results on KITTI-360.** As shown in Tab. 2, our model achieves 66.8% mIoU, outperforming all competing fusion methods. In particular, PMF Zhuang et al. (2021), TransFuser Prakash et al. (2021), TokenFusion Wang et al. (2022), and HRFuser Broedermann et al. (2023) lag significantly behind. Moreover, our approach surpasses CMX Liu et al. (2022) (64.3%) and CMNeXt Zhang et al. (2023) (65.3%), further demonstrating the effectiveness of the proposed fusion mechanism. These results highlight the superiority of our method for RGB-LiDAR segmentation, particularly in large-scale outdoor scenes.

**Results on DDD17 and DSEC.** Tab. 3 summarizes the results on the DDD17 and DSEC datasets. Our model achieves 77.85% mIoU and 97.10% Pixel Acc on DDD17 dataset and 74.77% mIoU and

Table 4: Semantic segmentation results on the DELIVER RGB-Event dataset.

| Method | Venue | Modal | Resolution | mIoU (%) |
|---|---|---|---|---|
| TokenFusion Wang et al. (2022) | CVPR'22 | RGB+Event | 1024×1024 | 45.6 |
| HRFuser Broedermann et al. (2023) | ITSC'23 | RGB+Event | 1024×1024 | 42.2 |
| CMX Liu et al. (2022) | TITS'23 | RGB+Event | 1024×1024 | 56.5 |
| CMNeXt Zhang et al. (2023) | CVPR'23 | RGB+Event | 1024×1024 | 57.5 |
| Any2Seg Zheng et al. (2024a) | ECCV'24 | RGB+Event | 512×512 | 57.8 |
| KTBNet Cai et al. (2025) | CVPR'25 | RGB+Event | - | 58.4 |
| Ours | - | RGB+Event | 512×512 | **60.3** |

Table 5: Semantic segmentation results on the UrbanLF-real and UrbanLF-syn datasets. "LF80", "LF33", and "LF8" denote using 80, 33, and 8 sub-aperture views as auxiliary modalities.

| Method | Venue | Modal | Real mIoU (%) | Syn mIoU (%) |
|---|---|---|---|---|
| SegFormer Xie et al. (2021) | NeurIPS'21 | RGB | 82.20 | 78.53 |
| OCR Yuan et al. (2020) | TCSVT'22 | RGB | 78.60 | 79.36 |
| DAVSS Zhuang et al. (2020) | TCSVT'20 | Video | 75.91 | 74.27 |
| TMANet Wang et al. (2021) | ICIP' 21 | Video | 77.14 | 76.41 |
| SA-Gate Chen et al. (2020b) | ECCV' 20 | RGB-D | - | 79.53 |
| PSPNet-LF Zhao et al. (2017) | CVPR'17 | RGB+LF33 | 78.10 | 77.88 |
| OCR-LF Sheng et al. (2022) | TCSVT'22 | RGB+LF33 | 79.32 | 80.43 |
| CMNeXt Zhang et al. (2023) | CVPR'23 | RGB+LF8 | 83.22 | 80.74 |
| CMNeXt Zhang et al. (2023) | CVPR'23 | RGB+LF33 | 82.62 | 80.98 |
| CMNeXt Zhang et al. (2023) | CVPR'23 | RGB+LF80 | 83.11 | 81.02 |
| Ours | - | RGB+LF8 | 82.59 | 80.98 |
| Ours | - | RGB+LF33 | 83.30 | 81.22 |
| Ours | - | RGB+LF80 | **83.99** | **81.91** |

95.60% Pixel Acc on DSEC dataset. Compared with the second-best RGB+Event fusion method, EISNet Xie et al. (2024), our approach consistently outperforms it in both mIoU and Pixel Acc. In contrast, event-only models such as ESEG-L Zhao et al. (2025), KWYAF Li et al. (2025b), and OpenESS Kong et al. (2024) achieve considerably lower scores, underscoring the importance of multi-modal fusion. These results validate the effectiveness of our method and set a new SOTA on the DDD17 and DSEC datasets.

**Results on DELIVER.** We further evaluate our approach on the DELIVER dataset, which provides high-resolution ($1024 \times 1024$) RGB-event pairs for semantic segmentation. Owing to GPU memory limitations, we train and evaluate our model on resized inputs of $512 \times 512$. As shown in Tab. 4, our method achieves 60.3% mIoU. Notably, even at half the original resolution, it surpasses recent SOTA methods such as KTBNet Cai et al. (2025) (58.4%), Any2Seg Zheng et al. (2024a) (57.8%), and CMNeXt Zhang et al. (2023) (57.5%). These results underscore the robustness of our fusion framework, even under reduced input resolution.

**Results on UrbanLF.** As shown in Tab. 5, our model achieves 83.99% mIoU on UrbanLF-real and 81.91% mIoU on UrbanLF-syn, outperforming CMNeXt Zhang et al. (2023) (83.22%/80.74%). Even under different sub-aperture configurations, our framework maintains strong performance, achieving 83.30%/81.22% with LF33 and 82.59%/80.98% with LF8. Compared with RGB-only and video-based methods, our approach delivers substantial improvements, underscoring the advantage of integrating sub-aperture views with RGB guidance.

Table 6: Semantic segmentation results on the MCubeS dataset with different modality combinations. "A" denotes AoLP, "D" denotes DoLP, and "N" denotes NIR.

| Method | Venue | Modal | mIoU (%) |
|---|---|---|---|
| MCubeSNet Liang et al. (2022) | CVPR'22 | RGB+A | 39.10 |
| CMNeXt Zhang et al. (2023) | CVPR'23 | RGB+A | 48.42 |
| Ours | - | RGB+A | **50.09** |
| MCubeSNet Liang et al. (2022) | CVPR'22 | RGB+A+D | 42.00 |
| CMNeXt Zhang et al. (2023) | CVPR'23 | RGB+A+D | 49.48 |
| Ours | - | RGB+A+D | **51.14** |
| MMTM Joze et al. (2020) | CVPR'20 | RGB+A+D+N | 39.71 |
| DRConv Chen et al. (2021) | CVPR'21 | RGB+A+D+N | 34.63 |
| DDF Zhou et al. (2021a) | CVPR'21 | RGB+A+D+N | 36.16 |
| TransFuser Prakash et al. (2021) | CVPR'21 | RGB+A+D+N | 37.66 |
| MCubeSNet Liang et al. (2022) | CVPR'22 | RGB+A+D+N | 42.86 |
| CMNeXt Zhang et al. (2023) | CVPR'23 | RGB+A+D+N | 51.54 |
| Ours | - | RGB+A+D+N | **52.22** |

**Results on MCubeS.** Tab. 6 presents the results on the MCubeS dataset. Using all four modalities (RGB+A+D+N), our method outperforms all compared approaches, including CMNeXt Zhang et al. (2023) and MCubeSNet Liang et al. (2022). Experiments with different modality combinations further demonstrate that each additional modality (A, D, and N) contributes to improved segmentation performance. Overall, fusing all four modalities yields the best results. The results for the ZJU dataset are provided in the **Appendix**.

## 4.1 ABLATION STUDIES

Tab. 7 summarizes the ablation results on the MFNet, DSEC, and DDD17 datasets, highlighting the contributions of each proposed component in terms of mIoU. Models are trained for 60 epochs on MFNet and DSEC, and 40 epochs on DDD17. Specifically, introducing BPLCA yields substantial gains, with a **+6.45% mIoU** improvement on MFNet, demonstrating that BPLCA effectively enhances cross-modal interaction.

Table 7: Ablation study of different architectures on the MFNet, DSEC, and DDD17 datasets.

| Architecture | MFNet mIoU (%) | DSEC mIoU (%) | DDD17 mIoU (%) |
|---|---|---|---|
| Baseline | 67.21 | 74.26 | 76.91 |
| + BPLCA | 73.66 | 74.04 | 77.57 |
| + BPLCA + DFCC | 76.36 | 74.29 | 77.73 |
| + BPLCA + $\mathcal{L}_{stage}$ | 76.57 | 74.37 | 77.63 |
| + BPLCA + DFCC + $\mathcal{L}_{stage}$ | **76.74** | **74.77** | **77.85** |

Incorporating DFCC further boosts performance—particularly on MFNet (76.36%)—by refining feature consistency. The dense stage-wise loss $\mathcal{L}_{stage}$ provides additional supervision and stabilizes optimization, leading to further improvements across all datasets. Finally, combining BPLCA, DFCC, and $\mathcal{L}_{stage}$ achieves the best performance: 76.74% mIoU on MFNet, 74.77% on DSEC, and 77.85% on DDD17. Overall, these results confirm that the three components are complementary: BPLCA strengthens modality fusion, DFCC enforces feature consistency, and $\mathcal{L}_{stage}$ supplies multi-level supervision—together delivering significant segmentation improvements.

## 5 CONCLUSION

In this work, we presented a multi-modal semantic segmentation framework that flexibly integrates RGB with diverse auxiliary modalities, including thermal, LiDAR, event, light field, and polarization data. Our approach features a selective fusion mechanism that dynamically activates the most informative auxiliary modality at each spatial location, a bidirectional polarity-aware linear cross-attention (BPLCA) combined with a dual feature consistency constraint (DFCC) for feature-aligned fusion, and a stage-wise supervision loss that progressively enforces cross-modal consistency. Extensive experiments on eight public benchmarks demonstrate that our method consistently outperforms SOTA approaches.

## ETHICS STATEMENT

This research adheres to the ethical standards of the ICLR community. The datasets used in our experiments are publicly available and widely adopted in prior works and do not contain personally identifiable or sensitive information. Our models are developed solely for academic research purposes. We acknowledge that semantic segmentation and multimodal perception techniques may potentially be applied in sensitive domains (e.g., surveillance, autonomous driving), and we encourage responsible use of our methods.

## REPRODUCIBILITY STATEMENT

To ensure reproducibility, we provide detailed descriptions of our model architecture, training settings, and experimental protocols in the paper. All hyperparameters and loss functions are explicitly specified. Furthermore, we will release the source code, pretrained models, and instructions for reproducing the reported results upon publication. This allows other researchers to validate our findings and extend our work.

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

# A APPENDIX

## A.1 USE OF LLMS

We used ChatGPT (OpenAI) solely for grammar polishing and improving readability in the introduction and related work sections. No content generation, data analysis, or result creation was performed using large language models (LLMs).

## A.2 DATASETS

An overview of the datasets used in our experiments is provided in Tab. 8.

Table 8: Summary of multi-modal datasets used in our experiments.

| Dataset | Modality | Classes | Train / Test | Resolution |
|---------|----------|---------|--------------|------------|
| MFNet Ha et al. (2017) | RGB-Thermal | 8 | 784 / 392 | $480 \times 640$ |
| KITTI-360 Liao et al. (2022) | RGB-LiDAR | 19 | 49004 / 12276 | $1408 \times 376$ |
| DDD17 Binas et al. (2017) | RGB-Event | 5 | 15950 / 3890 | $200 \times 346$ |
| DSEC Gehrig et al. (2021) | RGB-Event | 11 | 8082 / 2809 | $440 \times 640$ |
| DELIVER Zhang et al. (2023) | RGB-Event-Depth | 25 | 3893 / 1897 | $1024 \times 1024$ |
| UrbanLF Sheng et al. (2022) | RGB-Light Field | 14 | 580 / 80 (Real), 172 / 28 (Syn) | $623 \times 432, 640 \times 480$ |
| MCubeS Liang et al. (2022) | RGB-AoLP-DoLP-NIR | 20 | 302 / 102 | $1224 \times 1024$ |
| ZJU Xiang et al. (2021) | RGB-Polarization | 8 | 344 / 50 | $1224 \times 1024$ |

**RGB-T MFNet** Ha et al. (2017) is a driving-scene dataset with 1,569 aligned RGB-thermal pairs across 8 semantic classes. It provides 784/392/392 images for training, validation, and testing, respectively, covering both daytime and nighttime conditions.

**RGB-L KITTI-360** Liao et al. (2022) is a suburban driving dataset, providing 49,004 training and 12,276 validation images with a resolution of $1408 \times 376$. Following the Cityscapes dataset Cordts et al. (2016), it defines 19 semantic classes.

**RGB-E DDD17** Binas et al. (2017) includes over 12 hours of driving data collected with a DAVIS346B sensor, providing event streams and grayscale images at $200 \times 346$ resolution. We generate pseudo-labels using EV-SegNet Alonso & Murillo (2019), resulting in 5 semantic classes and splits of 15,950/3,890 samples.

**RGB-E DSEC** Gehrig et al. (2021) contains over 10k RGB-event frames with 11 categories, captured from stereo cameras in urban and rural environments. We follow Xie et al. (2024) for preprocessing and dataset splits (8082/2809 for training/testing).

**RGB-X DELIVER** Zhang et al. (2023) is a large-scale synthetic benchmark built in the CARLA simulator, supporting multi-modal segmentation across RGB, depth, event, and LiDAR streams. It contains over 47k annotated frames from six camera views under diverse conditions, including challenging weather (fog, rain, night) and five sensor degradation types (e.g., motion blur, exposure imbalance, LiDAR jitter). The dataset defines 25 semantic classes covering urban elements such as vehicles, roads, pedestrians, and vegetation.

**RGB-LF UrbanLF** Sheng et al. (2022) is a light-field dataset for urban-scene segmentation, including both real (580/80/164) and synthetic (172/28/50) subsets, each with 14 semantic classes. Central-view annotations are used for supervision.

**RGB-P MCubeS** Liang et al. (2022) consists of 500 samples with aligned RGB, NIR, and polarization cues (DoLP, AoLP), annotated with 20 material categories. The dataset is split into 302/96/102 for training, validation, and testing.

**RGB-P ZJU** Xiang et al. (2021) is a polarization semantic segmentation dataset collected in outdoor scenes. Each sample contains four polarized RGB images at $0°$, $45°$, $90°$, and $135°$, from which AoLP and DoLP are derived using Stokes parameters. It provides 344 training and 50 validation samples across 8 semantic classes, with image resolution of $1224 \times 1024$.

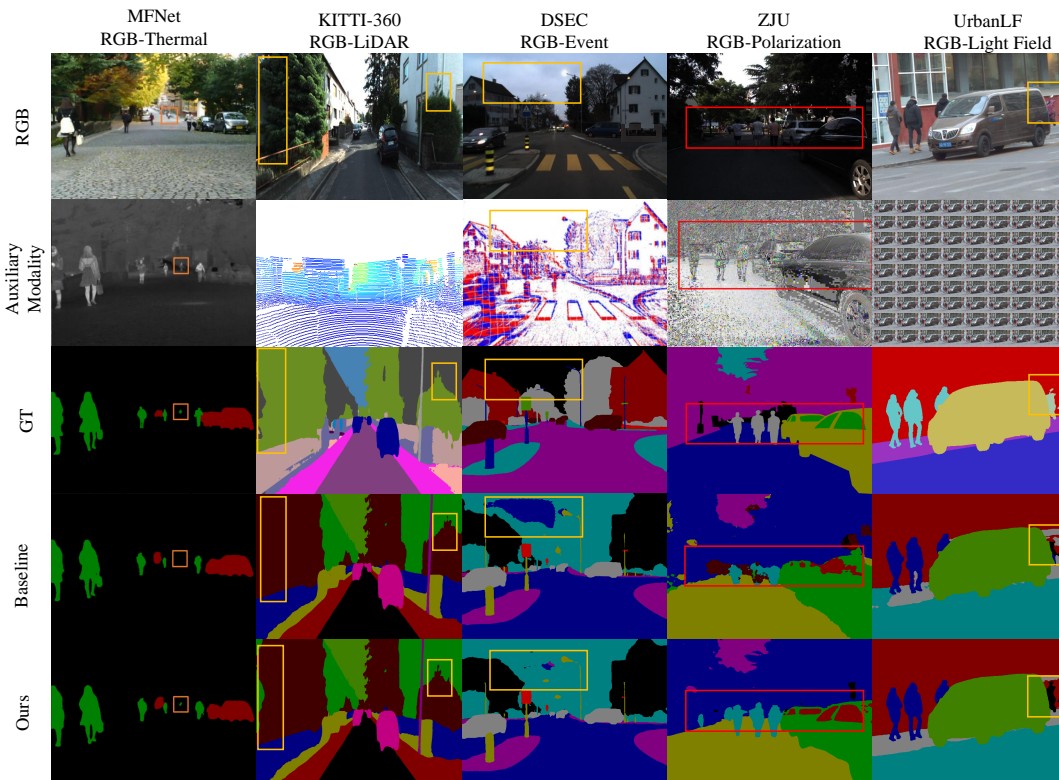

Figure 6: Qualitative results across multiple modalities. Our model consistently produces sharper boundaries and more accurate object recognition under challenging conditions.

## A.3 IMPLEMENTATION DETAILS

All models are trained on a single NVIDIA RTX 3090 GPU with a batch size of 2. We adopt AdamW Loshchilov & Hutter (2019) as the optimizer, with an initial learning rate of $6 \times 10^{-5}$, momentum of 0.9, weight decay of 0.01, and epsilon of $1 \times 10^{-8}$. The learning rate is scheduled using polynomial decay with a power of 0.9, preceded by a 10-epoch linear warm-up. Input images are resized to dataset-specific resolutions. Moreover, standard data augmentations are applied, including random resizing within [0.5, 2.0], horizontal flipping, color jitter, Gaussian blur, and random cropping.

## A.4 QUALITATIVE ANALYSIS

In Fig. 6, we present qualitative comparisons on five representative datasets: MFNet Ha et al. (2017) (RGB-Thermal), KITTI-360 Liao et al. (2022) (RGB-LiDAR), DSEC Gehrig et al. (2021) (RGB-Event), ZJU-RGBP Xiang et al. (2021) (RGB-Polarization), and UrbanLF-real Sheng et al. (2022) (RGB-LightField). On MFNet, our method more clearly distinguishes pedestrians—particularly small and distant ones—that the baseline often fails to separate from the background. On KITTI-360, it delineates roadside trees with sharper boundaries and smoother transitions between vegetation and surrounding regions. On DSEC, our model suppresses spurious responses in the background sky, yielding more stable predictions under fast-motion conditions. On ZJU-RGBP, improvements are evident around reflective surfaces, where pedestrians and vehicle windows are predicted with higher accuracy and fewer artifacts. On UrbanLF-real, our approach provides more consistent segmentation of riders, pedestrians, and background elements, maintaining coherent parsing even under occlusion. Across all benchmarks, these qualitative visualizations show that our fusion strategy consistently produces more reliable results, underscoring its superiority across diverse sensing scenarios.

Table 9: Semantic segmentation results on the ZJU RGB-P dataset. "P(A+D)" denotes using both AoLP and DoLP as auxiliary input.

| Method | Venue | Backbone | Modal | mIoU (%) |
|---|---|---|---|---|
| SwiftNet Orsic et al. (2019) | CVPR'19 | - | RGB | 80.3 |
| CMX Liu et al. (2022) | TITS'23 | MiT-B4 | RGB+A | 92.6 |
| Ours | - | MiT-B4 | RGB+A | **92.8** |
| CMX Liu et al. (2022) | TITS'23 | MiT-B4 | RGB+D | 92.5 |
| Ours | - | MiT-B4 | RGB+D | **93.1** |
| NLFNet Yan et al. (2021) | ROBIO'21 | - | RGB+P(A+D) | 84.4 |
| EAFNet Xiang et al. (2021) | OE'21 | - | RGB+P(A+D) | 85.7 |
| RoadFormer+ Huang et al. (2024) | TIV'24 | - | RGB+P(A+D) | 93.0 |
| Ours | - | MiT-B4 | RGB+P(A+D) | **93.8** |

## A.5 FURTHER RESULTS ANALYSIS

**Results on ZJU.** On the ZJU dataset, our method achieves 93.8% mIoU with both A and D inputs, surpassing all competing methods (see Tab. 9). Under different modality settings, it attains 92.8% with RGB+A and 93.1% with RGB+D, outperforming CMX Liu et al. (2022) in both cases. These results highlight the complementary role of polarization cues and demonstrate the effectiveness of our fusion strategy.

## A.6 LIMITATIONS AND FUTURE WORK

Despite these promising results, several limitations remain. The computational overhead of transformer-based dual encoders is still significant, especially at high resolutions or when handling multiple modalities simultaneously. Future work may explore lightweight backbones, efficient attention mechanisms, or model compression techniques to improve scalability.

