# OpenReview forum: "Multimodal Fusion of RGB and Complementary Modalities for Semantic Segmentation"
_ICLR.cc/2026/Conference — ICLR 2026 Conference Withdrawn Submission_

### Official Review · Reviewer_dFie · 2025-10-16

**Soundness:** 3
**Presentation:** 3
**Contribution:** 3
**Rating:** 4
**Confidence:** 5

**Summary:**

This paper proposed a bidirectional polarity-aware cross-modality fusion method that effectively captures complementary cues while enhancing feature alignment. The proposed method is evaluated on multiple benchmarks and achieves large gains.

**Strengths:**

1. The proposed solution is verified to be effective on different RGB-X segmentation benchmarks.
2. The proposed solution achieves large accuracy gains on the MFNet dataset.
3. Extensive ablation studies and analyses are presented.

**Weaknesses:**

1. It would be nice to compare the proposed fusion module against a list of existing fusion modules to better illustrate its superiority and computational efficiency.
2. The ablation study helps to show the gains of the architecture on MFNet. It would be nice to add more qualitative or interpretable analyses (e.g., feature visualizations, t-SNE visualizations) to help better understand the effectiveness of the proposed method. Why did existing methods fail to achieve high performance on MFNet? Why does BPLCA have a great benefit for RGB-T segmentation? These behaviors should be discussed and analyzed in detail.
3. More recent state-of-the-art general multimodal segmentation models like OmniSegmenter, MemorySAM, and AnySeg could be compared and discussed.

**Questions:**

1. How about RGB-D segmentation? This could be discussed, as RGB-D segmentation is one of the main benchmarks in RGB-X vision.
2. Would you consider presenting the detailed results in different modality-degradation scenarios on the DeLiVER benchmark?
3. Would you consider some experiments using different backbones? This would better verify the generalization capacity of your method.

---

### Official Review · Reviewer_k5UA · 2025-10-24

**Soundness:** 2
**Presentation:** 3
**Contribution:** 2
**Rating:** 4
**Confidence:** 4

**Summary:**

This paper proposes a framework for multi-modal semantic segmentation, designed to fuse RGB data with various complementary modalities. The central contribution is the Bidirectional Polarity-aware Cross-modality Fusion (BPCF) module, which aims to address feature misalignment and effectively integrate complementary information from heterogeneous sensors. The BPCF module consists of two main components: a Bidirectional Polarity-aware Linear Cross-Attention (BPLCA) mechanism to capture both positively and negatively correlated signals, and a Dual Feature Consistency Constraint (DFCC) with a stage-wise loss to enforce feature alignment. The overall architecture is a dual-encoder system that uses ScoreNet to dynamically select the most informative auxiliary modality at the patch level. The authors conduct extensive experiments across eight public datasets and five modality pairings to validate the method.

**Strengths:**

1. The paper is well organized and easy to follow.
2. The proposed method is evaluated on eight distinct datasets covering five different RGB+X combinations, supporting the framework's versatility and generalizability.

**Weaknesses:**

1. The overall framework heavily relies on prior works like CMNeXt, adopting its dual-encoder structure. The main novelty is confined to the BPCF module. However, the BPCF module itself appears to be a complex amalgamation of existing techniques without a clear, simple motivation for why this specific combination is necessary.
2. The use of ScoreNet for patch-level modality selection is a coarse approach. The fixed patch grid does not align with semantic content, potentially splitting a single object and forcing an arbitrary modality choice for its constituent parts.
3. The paper reports a remarkable 17% absolute mIoU improvement on the MFNet dataset, but the gains on other datasets are far more modest. Whether the method is over-tuned to this dataset? There should be a detailed analysis explaining this phenomenon.

**Questions:**

see weakness

---

### Official Review · Reviewer_groU · 2025-11-01

**Soundness:** 3
**Presentation:** 3
**Contribution:** 2
**Rating:** 4
**Confidence:** 5

**Summary:**

This paper presents a method for multimodal semantic segmentation that combines RGB images with other complementary sensing modalities, including thermal, LiDAR, event, polarization, and light-field data. The main technical contribution is a bidirectional polarity-aware cross-modality fusion (BPCF) module, which includes a polarity-aware linear cross-attention (PLCA) mechanism and a dual feature consistency constraint (DFCC). The authors evaluate their approach on eight different benchmarks covering five auxiliary modalities, and report strong performance, often achieving state-of-the-art results.

**Strengths:**

1. Figures throughout the paper (e.g., Fig. 1, 2, 3, 4, 5) provide clear visualizations of results, architecture, and information flow, aiding in the comprehension of nontrivial design choices.
2. This paper is easy to follow.

**Weaknesses:**

1. The proposed polarity-aware attention mechanism (Eq. 11, Page 5) adds extra computation compared to standard linear attention because of its explicit decomposition and gating operations. Although the authors claim that it operates in linear time and memory (Page 4), the paper does not provide any wall-clock runtime or memory usage analysis. This lack of practical efficiency evaluation could make it difficult to assess the method’s feasibility for resource-constrained deployment or for scenarios involving more than two modalities.
2. In Section 3.1 and Eqs. 1–2 (Page 3), ScoreNet is used to dynamically select the most informative patches from the auxiliary modalities. However, the paper provides little guidance on hyperparameter choices, threshold settings, or how ScoreNet handles ambiguous cases such as ties. It is also unclear how the method behaves if ScoreNet consistently suppresses a particular modality, which could lead to modal dominance or imbalance.
3. On the DELIVER dataset (Table 4), the method is evaluated at a reduced input resolution (512 × 512) due to resource constraints, whereas some competing methods use higher resolutions (1024 × 1024). While this limitation is acknowledged, the resulting score differences may be less meaningful. The paper also does not provide a detailed error analysis to quantify how this downscaling affects performance.
4.  The main novelty of the paper lies in the architectural design for feature fusion. However, the authors do not provide any analysis of latency, throughput, or real-time feasibility. This is particularly important given the use of Transformers and multiple input modalities, which are relevant for time-sensitive applications such as autonomous driving.
5. While ScoreNet (Eqs. 1–2) is justified as a mechanism for dynamically selecting the most informative patches, the paper provides little visualization or explanation of what features it actually selects in ambiguous or complex multi-modal regions. Including an analysis—such as heatmaps or selection frequency statistics—would strengthen the claims regarding the importance and effectiveness of adaptive modality selection.

**Questions:**

Please refer to the weaknesses section.

---

### Official Review · Reviewer_myaP · 2025-11-01

**Soundness:** 2
**Presentation:** 2
**Contribution:** 1
**Rating:** 2
**Confidence:** 5

**Summary:**

This paper aims to solve RGB-X semantic segmentation, where RGB is complemented by one of several modalities (thermal, LiDAR, event, polarization, light field). The authors propose a transformer-based, dual-encoder architecture with stage-wise fusion and introduce a Bidirectional Polarity-aware Cross-modality Fusion (BPCF) module. BPCF itself is made of (i) a Bidirectional Polarity-aware Linear Cross-Attention (BPLCA) that splits features into positive/negative parts and does cross-modal attention in both directions, and (ii) a Dual Feature Consistency Constraint (DFCC) that normalizes the two branches and enforces a stage-wise consistency loss. The proposed network is evaluated on five modality parings (RGB+T, RGB+Event, RGB+Lidar, RGB+Polarization, RGB+Light Field) on eight public datasets (MFNet, KITTI-360, DDD17, DSEC, DELIVER, UrbanLF-real/syn, MCubeS, ZJU). it shows performance improvements over CMX, CMNeXt, and a few recent modality-agnostic methods.

**Strengths:**

1. Unified evaluation across many RGB+X pairs.
* Testing on RGB+Thermal, RGB+Event, RGB+LiDAR, RGB+Polarization, and RGB+Light Field in one framework is valuable. Because many existing fusion papers only show one or two pairs. This makes the claim of “modality-agnostic” a bit more convincing.

**Weaknesses:**

1. Novelty on fusion architecture over existing RGB+X works is weak.
* The core claim is that BPCF/BPLCA “captures complementary cues and aligns heterogeneous features.” But this idea is generally shared over all RGB+X frameworks, such as CMX (cross-modal rectification + fusion), CMNeXt (self-query hub + parallel pooling), and Any2Seg (modality-agnostic fusion). They have two branches and cross-attention in both directions, and a feature-alignment/rectification step. Here, “polarity-aware” is mostly a kernelized linear attention with positive/negative decomposition on top of it. That is an incremental change, not a qualitatively new fusion paradigm. This is still a “dual encoder + cross-attention + consistency” network, like many RGB-X papers in 2022–2025.

2 Performance gains are not significant enough to be the contribution.
* The authors emphasize the MFNet jump (≈59.9 → 76.9 mIoU) as a “17% absolute” gain. But that is the best case. On other datasets, the performance improvements are quite marginal, making it difficult to consider the performance improvement itself a standalone contribution.

**Questions:**

Please see weakness

---

### Note · Authors · 2025-11-14

I have read and agree with the venue's withdrawal policy on behalf of myself and my co-authors.